# Levelling the Translational Gap for Animal to Human Efficacy Data

**DOI:** 10.3390/ani10071199

**Published:** 2020-07-15

**Authors:** Guilherme S. Ferreira, Désirée H. Veening-Griffioen, Wouter P. C. Boon, Ellen H. M. Moors, Peter J. K. van Meer

**Affiliations:** 1Department of Pharmaceutics, Utrecht Institute for Pharmaceutical Sciences, Utrecht University, 3512 JE Utrecht, The Netherlands; d.h.veening-griffioen@uu.nl (D.H.V.-G.); p.v.meer@cbg-meb.nl (P.J.K.v.M.); 2Copernicus Institute of Sustainable Development, Innovation Studies, Utrecht University, 3512 JE Utrecht, The Netherlands; w.p.c.boon@uu.nl (W.P.C.B.); e.h.m.moors@uu.nl (E.H.M.M.); 3Medicines Evaluation Board, 3531 AH Utrecht, The Netherlands

**Keywords:** animal model, drug development, translational research, FIMD, validation, systematic review, meta-analysis, investigator’s brochure, external validity

## Abstract

**Simple Summary:**

The value of animal research has been increasingly debated, owing to high failure rates in drug development. A potential contributor to this situation is the lack of predictivity of animal models of disease. Until recently, most initiatives focused on well-known problems related to the lack of reproducibility and poor reporting standards of animal studies. Only now, more methodologies are becoming available to evaluate how translatable the data from animal models of disease are. We discuss the use and implications of several methods and tools to assess the generalisability of animal data to humans. To analyse the relevance of animal research objectively, we must (1) guarantee the experiments are conducted and reported according to best practices; (2) ensure the selection of animal models is made with a clear and detailed translational rationale behind it. Once these conditions are met, the true value of the use of animals in drug development can be finally attested.

**Abstract:**

Reports of a reproducibility crisis combined with a high attrition rate in the pharmaceutical industry have put animal research increasingly under scrutiny in the past decade. Many researchers and the general public now question whether there is still a justification for conducting animal studies. While criticism of the current modus operandi in preclinical research is certainly warranted, the data on which these discussions are based are often unreliable. Several initiatives to address the internal validity and reporting quality of animal studies (e.g., Animals in Research: Reporting In Vivo Experiments (ARRIVE) and Planning Research and Experimental Procedures on Animals: Recommendations for Excellence (PREPARE) guidelines) have been introduced but seldom implemented. As for external validity, progress has been virtually absent. Nonetheless, the selection of optimal animal models of disease may prevent the conducting of clinical trials, based on unreliable preclinical data. Here, we discuss three contributions to tackle the evaluation of the predictive value of animal models of disease themselves. First, we developed the Framework to Identify Models of Disease (FIMD), the first step to standardise the assessment, validation and comparison of disease models. FIMD allows the identification of which aspects of the human disease are replicated in the animals, facilitating the selection of disease models more likely to predict human response. Second, we show an example of how systematic reviews and meta-analyses can provide another strategy to discriminate between disease models quantitatively. Third, we explore whether external validity is a factor in animal model selection in the Investigator’s Brochure (IB), and we use the IB-derisk tool to integrate preclinical pharmacokinetic and pharmacodynamic data in early clinical development. Through these contributions, we show how we can address external validity to evaluate the translatability and scientific value of animal models in drug development. However, while these methods have potential, it is the extent of their adoption by the scientific community that will define their impact. By promoting and adopting high quality study design and reporting, as well as a thorough assessment of the translatability of drug efficacy of animal models of disease, we will have robust data to challenge and improve the current animal research paradigm.

## 1. Introduction

Despite modest improvements, the attrition rate in the pharmaceutical industry remains high [1,2,3]. Although the explanation for such low success is multifactorial, the lack of translatability of animal research has been touted as a critical aspect [4,5]. Evidence of the problems of animal research’s modus operandi has been mounting for almost a decade. Research has shown preclinical studies have major design flaws (e.g., low power, irrelevant endpoints), are poorly reported, or both—leading to unreliable data which ultimately means that animals are subjected to unnecessary suffering and clinical trial participants are potentially placed at risk [6,7]. Increasing reports of failure to reproduce preclinical studies across several fields pointed to the need to increase current standards [8,9,10]. In these terms, the assessment of two essential properties of translational research—internal and external validity—is jeopardised [11].

Internal validity refers to whether the findings of an experiment in defined conditions are true [4]. Measures related to internal validity, such as randomisation and blinding, reduce or prevent several types of bias, and have a profound effect on study outcomes [12,13]. Initiatives, such as the Animals in Research: Reporting In Vivo Experiments (ARRIVE) and Planning Research and Experimental Procedures on Animals: Recommendations for Excellence (PREPARE) guidelines guidelines and harmonized animal research reporting principles (HARRP), address the issues related to the poor design and reporting of animal experiments [14,15,16]. While all of these initiatives can resolve most, if not all, issues surrounding internal validity, they are poorly implemented and their uptake by all stakeholders is remarkably slow [13,17].

If progress on the internal validity front has been insufficient, for external validity, it has been virtually absent. External validity is related to whether an experiment’s findings can be extrapolated to other circumstances (e.g., animal to human translation). For external validity, while the study design plays a significant role (e.g., relevant endpoints and time to treatment), there is another often overlooked dimension—the animal models themselves [18]. The pitfalls of using animals to simulate human conditions, such as different aetiology and lack of genetic heterogeneity, have been widely recognised for a long time [1,10,19,20]. Nonetheless, the few efforts to address external validity to the same extent as internal validity are still insufficient [21,22].

The results of a sizeable portion of animal studies are unreliable [8,9,23]. If we cannot fully trust the data generated by animal experiments, how can we assess their value? We argue that for the sensible evaluation of animal models of disease, we need to generate robust data first. To generate robust data, we need models that simulate the human disease features to the extent that allows for reliable translation between species. Through the selection of optimal animal models of disease, we can potentially prevent clinical trials from advancing based on data unlikely to translate. Our research focuses on the design of methods and tools to this end. In the following sections, we discuss the development, applications, limitations and implications of the Framework to Identify Models of Disease (FIMD), systematic reviews and meta-analysis and the IB-derisk to evaluate the external validity of animal models of disease.

## 2. The Framework to Identify Models of Disease (FIMD)

The evaluation of preclinical efficacy often employs animal models of disease. Here, we use ‘animal model of disease’ for any animal model that simulates a human condition (or symptom) for which a drug can be developed, including testing paradigms. While safety studies are tightly regulated, including standardisation of species and study designs, there is hardly any guidance available for efficacy [24]. New drugs often have new and unknown mechanisms of action, which require tailor-made approaches for their elucidation [24,25]. As such, it would be counterproductive for regulatory agencies and companies alike to predetermine models or designs for efficacy as it is done for safety. However, in practice, this lack of guidance has contributed to the performance of studies with considerable methodological flaws [6,26].

The assessment of the external validity of animal models has traditionally relied on the criteria of face, construct and predictive validity [27,28]. These concepts are generic and highly prone to user interpretation, leading to the analysis of disease models according to different disease parameters. This situation complicates an objective comparison between animal models. Newer approaches, such as the tool developed by Sams-Dodd and Denayer et al., can be applied to in vitro and in vivo models and consist of the simple scoring of five categories (species, disease simulation, face validity, complexity and predictivity) according to their proximity to the human condition [20,29]. Nevertheless, they still fail to capture relevant characteristics involved in the pathophysiology and drug response, such as histology and biomarkers.

To address the lack of standardisation and the necessity of a multidimensional appraisal of animal models of disease, we developed the Framework to Identify Models of Disease (FIMD) [21,22]. The first step in the development of FIMD was the identification of the core parameters used to validate disease models in the literature. Eight domains were identified: Epidemiology, Symptomatology and Natural History (SNH), Genetics, Biochemistry, Aetiology, Histology, Pharmacology and Endpoints. More than 60% of the papers included in our scoping review used three or fewer domains. As it stood, the validation of animal models followed the tendency of the field as a whole—no standardisation nor integration of the characteristics frequently mentioned as relevant.

Based on these results, we drafted questions about each domain to determine the similarity to the human condition (Table 1). The sheet containing the answers to these questions and references thereof is called a validation sheet. The weighting and scoring system weighs all domains equally. The final score can be visualised in a radar plot of the eight domains and, together with the validation sheet, facilitates the comparison of disease models at a high level. An example of a radar plot is presented in Figure 1. However, we know that the domains can be of different importance depending on the disease. For example, in a genetic disease, such as Duchenne Muscular Dystrophy (DMD), the genetic domain has a higher weight than in type 2 diabetes, in which environmental factors (e.g., diet) play a significant role.

At first, we also designed a weighting and scoring system based on the intended indication to compare models. This system divided diseases into three groups according to their aetiology—genetic, external and multifactorial. By establishing a hierarchical relationship between the domains according to a disease’s characteristics, we expected to allow a more sensitive scoring of the models. However, after reflection, indication-based scoring was not included for two reasons. The first reason was the added complexity of this feature, which would require an additional setup to be executed in a framework that is already information-dense, possibly limiting its implementation. The second reason was the lack of validation of said feature, since the numerical values were determined with a biological rationale but no mathematical/statistical basis. As such, any gain in score sensitivity would not necessarily indicate a difference between animal models. We settled for a generic system that weighted all domains equally until more data using this methodology are available. An example of a radar plot is presented in Figure 1.

To account for the low internal validity of animal research, we added a reporting quality and risk of bias assessment for the pharmacological validation section. This section includes all studies in which a drug intervention was tested. The reporting quality parameters were based on the ARRIVE guidelines, and the risk of bias questions were extracted from the tool published by the SYstematic Review Center for Laboratory animal Experimentation (SYRCLE) [14,30]. With this information, researchers can put pharmacological studies into context and evaluate how reliable the results are likely to be.

The final contribution of FIMD was a validation definition for animal models of disease. We grounded the definition of validation on the evidence provided for a model’s context of use, grading it into four levels of confidence. With this definition, we intentionally decoupled the connotation of a validated model being a predictive model. Rather, we reinforce that a validated animal model is a model with well-defined, reproducible characteristics.

To validate our framework, we first conducted a pilot study of two models of type 2 diabetes—the Zucker Diabetic Fatty (ZDF) rat and db/db mouse—chosen on the basis of their extensive use in preclinical studies. Next, we did a complete validation of two models of Duchenne Muscular Dystrophy (DMD)—the mdx mouse and the Golden Retriever Muscular Dystrophy (GRMD) dog. We chose the mdx mouse owing to its common use as a DMD model and the GRMD dog for its similarities to the human condition [31,32]. While only minor differences were found for the type 2 diabetes models, the models for DMD presented more striking dissimilarities. The GRMD dog scored higher in the Epidemiology, SNH and Histology domains, whereas the mdx mouse did so in the Pharmacology and Endpoints domains, the latter mainly driven by the absence of studies in dogs. Our findings indicate that the mdx mouse may not be appropriate to test disease-modifying drugs, despite its use in most animal studies in DMD [31]. If more pharmacology studies are published using the GRMD dog, it will result in a more refined assessment. A common finding in all the four models was the high prevalence of experiments for which the risk of bias could not be assessed.

We designed FIMD to avoid the limitations of previous approaches, which included the lack of standardisation and integration of internal and external validities. Nonetheless, it presents challenges of its own, ranging from the definition of disease parameters and absence of a statistical model to support a more sensitive weighting and scoring system to the use of publicly available (and often biased) data. The latter is especially relevant, as study design and reporting deficiencies of animal studies undoubtedly represent a challenge to interpret the resulting data correctly. While owing to these deficiencies, many studies are less informative; some may still offer potential insights if the data are interpreted adequately. We included the reporting quality and risk of bias assessment to force researchers to account for these shortcomings when interpreting the data.

FIMD integrates the key aspects of the human disease that an animal model must simulate. Naturally, no model is expected to mimic the human condition fully. However, understanding which features an animal model can and which it cannot replicate allows researchers to select optimal models for their research question. More importantly, it puts the results from animal studies into the broader context of human biology, potentially preventing the advancement of clinical trials based on data unlikely to translate.

## 3. Systematic Review and Meta-Analysis

Systematic reviews and meta-analyses of animal studies were one of the earliest tools to expose the status of the field [33]. Nonetheless, their application to compare animal models of disease is relatively recent. In FIMD’s pilot study, the two models of type 2 diabetes (the ZDF rat and db/db mouse) only presented slight differences in the general score. Since FIMD does not compare models quantitatively, we conducted a systematic review and meta-analysis to compare the effect of glucose-lowering agents approved for human use on the HbA1c [34]. We chose HbA1c as the outcome owing to its clinical relevance in type 2 diabetes drug development [35,36].

The results largely confirmed FIMD’s pilot study results—both models responded similarly to drugs, irrespective of the mechanism of action. The only exception was exenatide, which led to higher reductions in HbA1c in the ZDF rat. Both models predicted the direction of effect in humans for drugs with enough studies. Moreover, the quality assessment showed that animal studies are poorly reported: no study mentioned blinding at any level, and less than half reported randomisation. In this context, the risk of bias could not be reliably assessed.

The development of systematic reviews and meta-analyses to combine animal and human data offers an unprecedented opportunity to investigate the value of animal models of disease further. Nevertheless, translational meta-analyses are still uncommon [37]. A prospect for another application of systematic reviews and meta-analyses lies in comparing drug effect sizes in animals and humans directly. By calculating the degree of overlap of preclinical and clinical data, animal models could be ranked according to the extent they can predict effect sizes across different mechanisms of action and drug classes. The calculation of this ‘translational coefficient’ would include effective and ineffective drugs. Using human effect sizes as the denominator, a translational coefficient higher than 1 would indicate an overestimation of treatment effect, while a coefficient lower than 1, an underestimation. The systematisation of translational coefficients would lead to ‘translational tables’, giving additional insight on models’ translatability. These translational tables, allied with more qualitative approaches, such as FIMD, could form the basis for evidence-based animal model selection in the future.

Indeed, such a strategy is not without shortcomings. Owing to significant differences in the methodology of preclinical and clinical studies, such comparisons may present unusually large confidence intervals, complicating their interpretation. In addition, preclinical studies would need to match the standards of clinical research to a higher degree, including the use of more relevant endpoints that can be compared. Considerations on other design (e.g., dosing, route of administration), statistical (e.g., sample size, measures of spread) and biological matters (species differences) will be essential to develop a scientifically sound approach.

## 4. Can Animal Models Predict Human Pharmacologically Active Ranges? A First Glance into the Investigator’s Brochure

The decision to proceed to first-in-human trials is mostly based on the Investigator’s Brochure (IB), a document required by the Good Clinical Practice (GCP) guidelines [38]. The IB compiles all the necessary preclinical and clinical information for ethics committees and investigators to evaluate a drug’s suitability to be tested in humans. The preclinical efficacy and safety data in the IB are, thus, the basis for the risk–benefit analysis at this stage. Therefore, it is paramount that these experiments are performed to the highest standards to safeguard healthy volunteers and patients.

However, the results from Wieschoswki and colleagues show a different scenario [26]. They analysed 109 IBs presented for ethics review of three German institutional review boards. The results showed that the vast majority of preclinical efficacy studies did not report measures to prevent bias, such as blinding or randomisation. Furthermore, these preclinical studies were hardly ever published in peer-reviewed journals and were overwhelmingly positive—only 6% of the studies reported no significant effect. The authors concluded IBs do not provide enough high quality data to allow a proper risk–benefit evaluation of investigational products for first-in-human studies during the ethics review.

In an ongoing study to investigate the predictivity of the preclinical data provided in the Ibs, we are evaluating whether animal models can predict pharmacologically active ranges in humans. Since pharmacokinetic (PK) and pharmacodynamic (PD) data are often scattered throughout the IB across several species, doses and experiments, integrating it can be challenging. We are using the IB-derisk, a tool developed by the Centre for Human Drug Research (CHDR), to facilitate this analysis [39]. The IB-derisk consists of a colour-coded excel sheet or web application (www.ib-derisk.org), in which PK and PD data can be inputted. It allows the extra- and interpolation of missing PK parameters across animal experiments, facilitating dose selection in first-in-human trials. The IB-derisk yields yet another method to discriminate between animal models of disease. With sufficient data, the drug PK and PD from preclinical studies of a single model and clinical trials of correspondent drugs can be compared. This analysis, when combined with PK/PD modelling, can serve as a tool to select the most relevant animal model based on the mechanism of action and model characteristics. Preliminary (and unpublished) results suggest that animal models can often predict human pharmacologically active ranges. How the investigated pharmacokinetic parameters relate to indication, safety, and efficacy is still unclear.

To build on Wieschowski’s results, we have been collecting data on the internal validity and reporting quality of animal experiments. Our initial analysis indicates the included IBs also suffer from the same pitfalls identified by Wieschowski, suggesting that such problems are likely to be widespread. In addition, only a few IBs justified their choice of model(s) of disease, and none compared their model(s) to other options to better understand their pros and cons. This missing information is crucial to allow for risk–benefit analysis during the ethics review process.

## 5. Levelling the Translational Gap for Animal to Human Efficacy Data

Together, FIMD, systematic reviews and meta-analyses and the IB-derisk have the potential to circumvent many limitations of current preclinical efficacy research. FIMD allows the general characterisation of animal models of disease, enabling the identification of their strengths and weaknesses. The validation process results in a sheet and radar plot that can be compared easily. Because the validation is indication-specific, the same model may have different scores for different diseases, allowing for a more nuanced assessment. The addition of measures of the risk of bias and reporting quality guarantees researchers have enough information to scrutinise the efficacy data. The IB-derisk tool correlates human and animal PK and PD ranges, adding a quantitative measure of response to the pharmacological validation of FIMD. Finally, a systematic review and meta-analysis of efficacy studies can show whether the pharmacological effects in animals translate into actual clinical effects.

This extensive characterisation of disease models—and their disparities to patients—is the only way to account for species differences that may impact the translation [11,40]. The largest study so far to analyse the concordance between animal and human safety data supports this premise [41]. For example, while rabbits can accurately predict arrhythmia in humans, the Irwin test in rats, a required safety pharmacology assessment, is of limited value. Thus, we need to focus on understanding the animal pathophysiology to the degree that allows us to assess for what kind of research a disease model is suitable. Especially for complex multifactorial conditions (e.g., Alzheimer’s disease), the use of multiple models that simulate different aspects is likely to provide a more detailed and reliable picture [42]. For instance, Seok and colleagues have shown that human and mouse responses to inflammation vary significantly according to their aetiology [43]. While genomic responses in human trauma and burns were highly correlated (R^2^ = 0.91), human to mouse responses correlated poorly, with an R^2^ < 0.1. These considerations are fundamental to select a model of disease to assess the efficacy of potential treatments, and FIMD can serve as the tool to compile and integrate this information. However, by itself, FIMD cannot prevent poor models from being used. The validation of all existing models may result in low scores, and thus, the inability to identify a relevant model. While the researcher may still choose to pick one of these models for a specific reason, FIMD offers institutional review boards and funders a scientifically-sound rationale to refuse the performance of studies in poor animal models—even if it means no animal research should be conducted.

There are some constraints for the implementation of the methods and tools we described. All of them require significant human and financial investments. Nonetheless, the estimated cost of irreproducibility (~30 billion USD per year in the US alone) likely dwarfs the cost for the broad implementation of these strategies [44]. Besides, this initial investment can be mitigated over time. For instance, FIMD validation sheets could be available in peer-reviewed, open access publications that can be updated periodically. This availability will prevent researchers from different institutions from unnecessarily repeating studies. Additionally, subsequent FIMD updates will be much faster than the first validation. In the same vein, if the application of systematic reviews and meta-analyses and the IB-derisk become commonplace, training will become widespread and their execution will be more efficient. The development of automated methods that can compile vast amounts of data will certainly aid to this end [45,46].

Animal research is already a cost-intensive and often long endeavour. By conducting experiments with questionable validities, we are misusing animals—which is expressively prohibited by the European Union (EU) Directive 2010/63. Only with a joint effort involving researchers in academia and industry, ethics committees, funders, regulatory agencies and the pharmaceutical industry, we can improve the quality of animal research. By applying FIMD, systematic reviews and meta-analysis and the IB-derisk, researchers can identify more predictive disease models, potentially preventing clinical trials starting based on unreliable data. These approaches can be implemented in the short-term in both the academic and industrial settings since the training requires only a few months.

Concomitantly, the other stakeholders must create an environment that encourages the adoption of best practices. Ethics committees have a unique opportunity to incentivise higher standards, since an unfavourable assessment can prevent poorly designed experiments from even starting. However, they now frequently base their decisions on subpar efficacy data [26]. In addition, the lack of a detailed disease model justification often results in the selection of disease models based on tradition, rather than science [47]. The request of a more detailed translational rationale for each model choice (e.g., by requiring models are evaluated with FIMD), as well as the enforcement of reporting guidelines, can act as gatekeepers for flawed study designs and improve the risk–benefit analysis significantly [48].

Funders can require the use of systematic reviews and meta-analyses and a thorough assessment of the translational relevance of selected animal models (e.g., FIMD). They can facilitate the adoption of these measures by reserving some of their budget specifically for training and the implementation of these strategies. Over time, FIMD and systematic reviews and meta-analyses can eventually become an essential requirement to acquire funding. Provided there is a grace period of at least a year, funders could request a stricter justification of model selection as early as their next grant round.

Journal editors and reviewers must actively enforce reporting guidelines for ongoing submissions, as endorsing them does not improve compliance [49]. Promoting adherence to higher reporting standards will also preserve the journal’s reputation. In the medium-term, journals can provide a list with relevant parameters (e.g., randomisation, blinding) with line references to be filled out by the authors in the submission system. Such a system would facilitate reporting quality checks by editors and reviewers without increasing the review length substantially. Registered reports allow input on the study design since they are submitted before the beginning of experiments. Additionally, since acceptance is granted before the results are known, the registered reports also encourage the publication of negative results, acting against the publication bias for positive outcomes.

Regulatory agencies can shape the drug development landscape significantly with some key actions. For instance, updating the IB guidelines by requiring a more extensive translational rationale for each animal model employed would facilitate the risk–benefit analysis by assessors and ethics committees alike. This rationale should include not only an evaluation of the animal model itself but also how it compares to other available options. Furthermore, agencies could review disease-specific guidance to include a more comprehensive account of efficacy assessment by exploring the use of disease models in safety studies [24,50]. The simultaneous evaluation of efficacy and safety can result in more informative studies, which are more likely to translate to the clinic. Finally, the output of the translational assessments of animal models of disease can be incorporated within periodic updates of guidelines, for instance, as an extended version of Sheean and colleagues’ work [51]. Scientific advice can be used as a platform to discuss translational considerations early in development. By presenting the strengths and weaknesses of validated disease models, agencies can promote optimal model selection without precluding model optimisation and the development of new approaches.

Furthermore, drug development companies can significantly benefit from the implementation of these measures in the medium-term. Larger companies can perform a thorough assessment of preclinical data of internal and external assets using FIMD, systematic review and meta-analysis and the IB-derisk. At the same time, small and medium enterprises can provide data in these formats to support their development plan. Ultimately, the selection of more predictive disease models will lead to more successful clinical trials, increasing the benefit and reducing the risks to patients, and lower development costs.

A positive side-effect of these strategies is the increased scrutiny of design and animal model choices. Instead of a status-quo based on tradition and replication of poor practices, we can move forward to an inquisitive and evidence-based modus operandi [7]. This change of culture is sorely needed in both academic and industrial institutions [52]. A shift toward a stricter approach—more similar to clinical trials, from beginning to end—is warranted [52].

This shift should include the, possibly immediate, preregistration and publication of animal studies on a public online platform, such as preclinicaltrials.eu [6,7,53,54,55]. As with clinical trials, where the quality of the preregistration and the actual publication of results are still a concern, this is unlikely to be sufficient [56,57]. Higher standards of study design, performance and reporting are necessary to increase the quality of the experiments. Currently, Good Laboratory Practices (GLP) are only required for preclinical safety, but not efficacy assessment. However, GLP experiments can be costly, making them prohibitive for academia, where many animal studies are conducted [58]. A change toward the Good Research Practices (GRP), published by the World Health Organisation, offers a feasible alternative [59]. The GRP consists of several procedures to improve the robustness of data and reproducibility, similarly to the GLP, but less strict. With the combination of efficacy and safety studies in disease models, the application of GRP instead of GLP may also be implemented in academia. Together with the employment of internal (e.g., ARRIVE and PREPARE guidelines, HARRP) and external validity (e.g., FIMD, systematic review and meta-analysis, IB-derisk) tools, these proposals tackle critical shortcomings of current animal research. The harmonisation of requirements across stakeholders will be crucial for a successful change of mindset.

Our suggestions to improve the preliminary assessment of efficacy in animals are likely to face resistance from a considerable part of the scientific community [60]. Nevertheless, so did the introduction of actions to improve the quality of clinical trials [53]. The requirement for preregistration and the establishment of blinded and randomised clinical trials as the gold standard have improved clinical research substantially. The current requirements for clinical trials to be executed are also time-consuming, but few would doubt their importance. It is time we apply the same rigour to animal research [48,54,61].

At present, the discussions over the translatability of animal studies are often based on questionable data. If many preclinical studies are poorly conducted, we cannot expect their results to be translatable to the clinical context. Another overlooked point is the definition of translatability standards. Provided that studies are perfectly designed, employ optimal models and are conducted meticulously, what is the level of concordance that society and scientists are willing to accept? This lack of standardisation leads to the extraordinary situation of both proponents and opponents of the current animal research paradigm citing the same data to defend their arguments [41,62,63].

This reflection is especially relevant for the discussion of alternatives to animal studies. Many in silico and in vitro (particularly organoids and organs-on-a-chip) approaches are in development, aiming to replace animal use in drug development partially or entirely, but they also have limitations [50,64]. For instance, organoids cannot simulate organ–organ interactions, and the organ-on-a-chip technology has not resolved the lack of a universal medium to connect different organs as well as its inability to replicate the immune response, endocrine system or gut-microbiome [65]. If these systems have the edge over animal studies owing to their human origin, they do not reproduce the known and unknown interactions of a whole intact organism. Additionally, they encounter challenges related to validation, reproducibility and reporting, akin to their animal counterparts [66,67,68]. Hence, in their present form, these approaches are better applied as a complement, rather than as a replacement, to animal research.

In the foreseeable future, animal research is unlikely to be eliminated. The criticism of the present state of affairs of preclinical research is indeed justifiable. However, dismissing the value of animals on the grounds of questionable data seems excessive. Meanwhile, our efforts must be focused on improving the robustness of animal data generated now. We already have tools available to address most, if not all, internal and external validity concerns. Only a thorough assessment of higher quality animal data will determine whether animal research is still a valid paradigm in drug development.

## 6. Final Considerations

As it stands, the scepticism over the justification of animal research is well-founded. The data on which we routinely base our decisions in drug development are often unreliable. A significant reappraisal of the current standards for animal research is warranted. We must design, conduct and report preclinical studies with the same rigour as clinical studies. Moreover, we must scrutinise animal models of disease to ensure they are relevant to the evaluation of preliminary efficacy or any other question we are asking. If there are no relevant animal models of disease, then preclinical testing in other platforms must be pursued.

Those changes will undoubtedly have a cost. Nevertheless, all stakeholders must be willing to invest human and financial resources to drive a change in culture and practice. Only when we promote and adopt high quality study design and reporting as well as a thorough assessment of animal models’ translatability, will we have robust data to challenge and improve the current paradigm.

## Figures and Tables

**Figure 1 animals-10-01199-f001:**
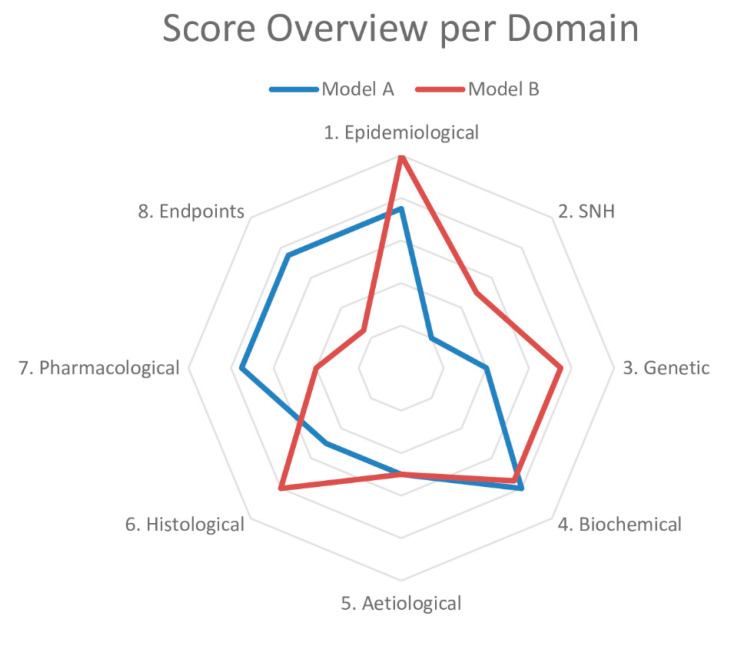
Example of a radar plot obtained with the validation of two animal models using the Framework to Identify Models of Disease (FIMD). SNH—Symptomatology and Natural History. Extracted from Ferreira et al. [21,22].

**Table 1 animals-10-01199-t001:** Questions per domain with weighting per questions. Extracted from Ferreira et al. [21,22].

	Weight
1. EPIDEMIOLOGICAL VALIDATION	12.5
1.1 Is the model able to simulate the disease in the relevant sexes?	6.25
1.2 Is the model able to simulate the disease in the relevant age groups (e.g., juvenile, adult or ageing)?	6.25
2. SYMPTOMATOLOGY AND NATURAL HISTORY VALIDATION	12.5
2.1 Is the model able to replicate the symptoms and co-morbidities commonly present in this disease? If so, which ones?	2.5
2.2 Is the natural history of the disease similar to human’s regarding: 2.2.1 Time to onset	2.5
2.2.2 Disease progression	2.5
2.2.3 Duration of symptoms	2.5
2.2.4 Severity	2.5
3. GENETIC VALIDATION	12.5
3.1 Does this species also have orthologous genes and/or proteins involved in the human disease?	4.17
3.2 If so, are the relevant genetic mutations or alterations also present in the orthologous genes/proteins?	4.17
3.3 If so, is the expression of such orthologous genes and/or proteins similar to the human condition?	4.16
4. BIOCHEMICAL VALIDATION	12.5
4.1 If there are known pharmacodynamic (PD) biomarkers related to the pathophysiology of the disease, are they also present in the model?	3.125
4.2 Do these PD biomarkers behave similarly to humans’?	3.125
4.3 If there are known prognostic biomarkers related to the pathophysiology of the disease, are they also present in the model?	3.125
4.4 Do these prognostic biomarkers behave similarly to humans’?	3.125
5. AETIOLOGICAL VALIDATION	12.5
5.1 Is the aetiology of the disease similar to humans’?	12.5
6. HISTOLOGICAL VALIDATION	12.5
6.1 Do the histopathological structures in relevant tissues resemble the ones found in humans?	12.5
7. PHARMACOLOGICAL VALIDATION	12.5
7.1 Are effective drugs in humans also effective in this model?	4.17
7.2 Are ineffective drugs in humans also ineffective in this model?	4.17
7.3 Have drugs with different mechanisms of action and acting on different pathways been tested in this model? If so, which?	4.16
8. ENDPOINT VALIDATION	12.5
8.1 Are the endpoints used in preclinical studies the same or translatable to the clinical endpoints?	6.25
8.2 Are the methods used to assess preclinical endpoints comparable to the ones used to assess related clinical endpoints?	6.25

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
