# Peer review of "Levelling the Translational Gap for Animal to Human Efficacy Data"

_animals, 2020, doi:10.3390/ani10071199_

Round 1
Reviewer 1 Report
This manuscript addresses the very important issue of choosing the most appropriate pre-clinical animal models for translation into the clinical context. As such its subject matter is appropriate for Animals. Whilst I found the paper to have material with the merit for publication, I am of the opinion that it lacks sufficient focus in its current state that its important message is lost. For that reason, I recommend major review; major in terms of being edited significantly, but I do not think that it is a substantial task. I detail my concerns below.
Line 53: " …was already mounting" should be "has been mounting"
Lines 53-54: "Research has shown ... e.g. low power, irrelevant endpoints)." Surely this points to a far more profound issue than reporting standards? Such experiments should not be allowed to be performed because they are bound both to be poor science and to have by definition caused animal suffering for no benefit. As such, I would argue that this is a flaw of the ethical review process (ERP). I make this comment having been the chair of an institutional ERP and a lay member of the ERP of an internationally-renowned university. In my view, there may be some value in improving journal requirements for publication, but refusing publication of poorly constructed experiments doesn't stop the experiments.
Line 73: "sizable" should be "sizeable"
Line 86: "... any guidance is available ..." omit "is"
Lines 115-116: I would have liked to see some elaboration of these two issues without recourse to reading the original work. One could, for example, conclude that: "added complexity" might lead to little uptake because too little benefit is gained from the extra work involved; and "lack of validation" means that one cannot be certain of its worth at all.
Line 137: "mainly driven by the absence of studies in the dog". This lack should encourage experiments with the dog model to confirm whether or not it is a suitable model to be used.
Lines 142-145: "Nonetheless ...biased) data." Are these deficiencies surmountable?
Line 148: "to select optimal models". Of course, the best model available may still be a poor model. Given the issue raised at Lines 53-54, specifically experiments that are inherently flawed, can FIMD exclude models - even all the available models - to ensure unnecessary animal suffering is avoided?
Lines 149-151: This is a key message, and in my view should be emphasised in the introduction and abstract.
Lines 165-166: "that animal studies are poorly reported ...". In what way? Be more specific.
Lines 169-170: the authors appear to have used only one search term, "translational meta-analysis". The paucity of retrieved results might by attributable to only having used this particular term. With any relatively novel approach, terms do not achieve currency immediately.
Lines 175-178: In my opinion, the authors are drifting into the realm of calculating a novel index because it is possible without considering what the calculation produces. Without a measure of spread (e.g. standard error) the only acceptable coefficient would be exactly 1.
Lines 182-187: In short, this approach is a long time from giving tangible benefits.
Lines 216-217: I think that the message of this sentence is very important, but that it could be much clearer. Perhaps, the authors could consider something like: "... Wieschowski, suggesting that such problems are likely to be widespread."
Lines 257-258: This sentence is a non-controversial demand. However it is too bland, in my opinion, because it does not fit with current publication processes. For example, this journal requires referees to perform a review within a week, and referees have other activities as well as refereeing. I would recommend that editors should return manuscripts to authors if they lack the required supplementary information rather than sending them to referees in the first instance.
Lines 278-279: should read: "... within periodic updates of guidelines, ..."
Line 280: should be: "... the strengths and weaknesses of validated disease models, ..."
Lines 282- 319: This appears to be little more than a creditable wish-list. As such, I find it distracts from the message of the need to tighten the requirements for selection of models and reporting details. The authors should consider omitting much, if not all, of this paragraph.
Overall, I suggest the authors think very clearly about the message for change that can be achieved, and then structure the paper accordingly. At present, it contains far too much listing of changes that could be made without identifying either timescales (short- versus long-term) or detailing the specific bodies (e.g. my comment re editors rather than reviewers) who should be responsible for the changes and, as such, the impulse to improve will be blunted.
Author Response
Reviewer #1: We thank the reviewer for the thorough review of our paper. We address the reviewer comments point-by-point below:
- Line 53: “ …was already mounting” should be “has been mounting”
We have edited the text accordingly.
1.2 Lines 53-54: "Research has shown ... e.g. low power, irrelevant endpoints)." Surely this points to a far more profound issue than reporting standards? Such experiments should not be allowed to be performed because they are bound both to be poor science and to have by definition caused animal suffering for no benefit. As such, I would argue that this is a flaw of the ethical review process (ERP). I make this comment having been the chair of an institutional ERP and a lay member of the ERP of an internationally-renowned university. In my view, there may be some value in improving journal requirements for publication, but refusing publication of poorly constructed experiments doesn't stop the experiments.
We agree with the reviewer that journal editors and reviewers alone cannot improve animal research on their own. Indeed, the role of Institutional Review Boards (IRBs) in requiring higher standards for animal studies is paramount. IRBs have the unique opportunity to stop potentially unsound animal studies from even starting. A careful assessment of study design and translational rationale can lead to a considerable improvement of animal research. However, none of these stakeholders can drive change in the field on their own. We think that such a fundamental change of culture can only happen when all the relevant parties are on board. Scientific publication is still the main parameter through which researchers in the academic setting are assessed. If we consider a significant portion of animal studies is conducted in academia, pressure for higher research standards at the journal level can be a strong incentive for improvement. Thus, action in unison across researchers and institutions is necessary. We edited the text to reflect this position more clearly. It now reads “Ethics committees have a unique opportunity to incentivise higher standards since an unfavourable assessment can prevent poorly designed experiments from even starting. However, they now frequently base their decisions on subpar efficacy data [26]. Also, the lack of a detailed disease model justification often results in the selection of disease models based on tradition, rather than science [47]. The request of a more detailed translational rationale for each model choice (e.g. by requiring models are evaluated with FIMD), as well as the enforcement of reporting guidelines, can act as gatekeepers for flawed study designs and improve the risk-benefit analysis significantly [48].” L. 341-7
1.3 Line 73: "sizable" should be "sizeable"
We have edited the text accordingly.
1.4 Line 86: "... any guidance is available ..." omit "is"
We have edited the text accordingly.
1.5 Lines 115-116: I would have liked to see some elaboration of these two issues without recourse to reading the original work. One could, for example, conclude that: "added complexity" might lead to little uptake because too little benefit is gained from the extra work involved; and "lack of validation" means that one cannot be certain of its worth at all.
We edited the text accordingly. It now reads “However, the indication-based scoring was not included after the review process of the manuscript for two reasons. The first reason was the added complexity of this feature, which would require an additional setup to be executed in a framework that is already information-dense, possibly limiting its implementation. The second reason was the lack of validation of said feature since the numerical values were determined with a biological rationale but no mathematical/statistical basis. As such, any gain in score sensitivity would not necessarily indicate a difference between animal models. We settled for a generic system that weighted all domains equally until more data using this methodology are available. An example of a radar plot is presented in Figure 1.” L.130-8.
1.6 Line 137: "mainly driven by the absence of studies in the dog". This lack should encourage experiments with the dog model to confirm whether or not it is a suitable model to be used.
The weighting and scoring system we developed for FIMD weights the lack of studies with only 10% of the score. Based on our review of the literature, the GRMD dog model is under utilised – it lacked studies across several drugs included in the pharmacological and endpoints domains. This absence of studies resulted in the GRMD dog receiving lower scores than the mdx mouse in these domains. While we agree that to test whether the GRMD is a better model, more studies should be done, it is neither our goal nor intent to recommend studies for this purpose. We edited the text to reflect that the publication of studies in the GRMD model will result in a more refined FIMD assessment. Now it reads “The GRMD dog scored higher in the Epidemiology, SNH and Histology domains whereas the mdx mouse did so in the Pharmacology and Endpoints domains, mainly driven by the absence of studies in the dog. Our findings indicate the mdx mouse may not be appropriate to test disease-modifying drugs, despite its use in most animal studies in DMD [31]. If more pharmacology studies are published using the GRMD dog, it will result into a more refined assessment.” L.177-182
1.7 Lines 142-145: "Nonetheless ...biased) data." Are these deficiencies surmountable?
The study design and reporting deficiencies of animal studies undoubtedly represent a challenge to interpret the resulting data correctly. While owing to these deficiencies, many studies are less informative; some may still offer potential insights if the data are interpreted adequately. Assuming these shortcomings are insurmountable would mean dismissing a substantial learning opportunity. In clinical research, the conclusions from a randomised, double-blind trial are weighed differently than an open-label single-arm study. We need to apply the same standards to animal research to surmount the limitations of the current data. Concomitantly, we need to promote better study design, to generate more robust data and reporting quality, to differentiate between well and poorly designed studies accurately. These measures allow us to use available data while understanding and accounting for their deficiencies and encourage the implementation of better practices.
1.8 Line 148: "to select optimal models". Of course, the best model available may still be a poor model. Given the issue raised at Lines 53-54, specifically experiments that are inherently flawed, can FIMD exclude models - even all the available models - to ensure unnecessary animal suffering is avoided?
The reviewer is completely correct. We designed FIMD to guide the assessment, validation and comparison of animal models of disease. As such, it is possible that the validation of all the models results in low scores, and thus, the inability to identify a relevant model. While the researcher may still pick one of these models for whatever justification, FIMD also offers institutional review boards and funders a scientifically-sound rationale to refuse the performance of studies in poor animal models – even if it means no animal research should be conducted.
1.9 Lines 149-151: This is a key message, and in my view should be emphasised in the introduction and abstract.
We edited the abstract and the introduction to emphasise the message. It now reads “As for external validity, progress has been virtually absent. However, the selection of optimal animal models of disease potentially prevents clinical trials from advancing based on data unlikely to translate. Here, we discuss three contributions to tackle the evaluation of the predictive value of animal models of disease themselves.” L.31-2 and “To generate robust data, we need models that simulate the human disease features to the extent that allows a reliable translation between species. Through the selection of optimal animal models of disease, we can potentially prevent clinical trials from advancing on the basis of data unlikely to translate. Our research focuses on the design of methods and tools to this end.” L.78-81
1.10 Lines 165-166: "that animal studies are poorly reported ...". In what way? Be more specific.
We edited the text to improve clarity. It now reads: “Moreover, the quality assessment showed that animal studies are poorly reported: no study mentioned blinding at any level and less than half reported randomisation. In this context, the risk of bias could not be reliably assessed.” L. 217-9.
1.11 Lines 169-170: the authors appear to have used only one search term, "translational meta-analysis". The paucity of retrieved results might by attributable to only having used this particular term. With any relatively novel approach, terms do not achieve currency immediately.
We agree that the only search term used is not optimal. Our point is that translational meta-analyses are still uncommon, and the paper by Leenaars and colleagues demonstrates it clearly. We edited the text to improve clarity. It now reads “The development of systematic reviews and meta-analyses to combine animal and human data offers an unprecedented opportunity to investigate the value of animal models of disease further. Nevertheless, translational meta-analyses are still uncommon [37]. A prospect for another application of systematic reviews and meta-analyses lies in comparing drug effect sizes in animals and humans directly.” L.220-4.
1.12 Lines 175-178: In my opinion, the authors are drifting into the realm of calculating a novel index because it is possible without considering what the calculation produces. Without a measure of spread (e.g. standard error) the only acceptable coefficient would be exactly 1.
The reviewer has a point since a measure of spread would be fundamental in this index. We also discuss it in our limitation analysis, citing large confidence intervals as a concern for the development of translational tables. We edited the text to improve clarity. It now reads “Indeed, such a strategy is not without shortcomings. Owing to significant differences in the methodology of preclinical and clinical studies, such comparisons may present unusually large confidence intervals, complicating their interpretation. Also, preclinical studies would need to match the standards of clinical research to a higher degree, including the use of more relevant endpoints that can be compared. Considerations on other design (e.g. dosing, route of administration), statistical (e.g. sample size, measures of spread) and biological matters (species differences) will be essential to develop a scientifically sound approach.” L. 232-8
1.13 Lines 182-187: In short, this approach is a long time from giving tangible benefits.
We agree with the reviewer: the implementation of the proposed methodology will still require some time. However, this is the case for every methodology in development. One of our goals in this manuscript is to provide perspectives on tools and methodologies that can be used to discriminate among animal models.
1.14 Lines 216-217: I think that the message of this sentence is very important, but that it could be much clearer. Perhaps, the authors could consider something like: "... Wieschowski, suggesting that such problems are likely to be widespread."
We edited the text to improve clarity. It now reads “Our initial analysis indicates the included IBs also suffer from the same pitfalls identified by Wieschowski, suggesting that such problems are likely to be widespread.” L. 283-5
1.15 Lines 257-258: This sentence is a non-controversial demand. However it is too bland, in my opinion, because it does not fit with current publication processes. For example, this journal requires referees to perform a review within a week, and referees have other activities as well as refereeing. I would recommend that editors should return manuscripts to authors if they lack the required supplementary information rather than sending them to referees in the first instance.
We understand the reviewer concerns regarding our proposals. The current publication process offers challenges for editors and reviewers alike. Editors must make a first appraisal of manuscripts and should include reporting quality checks in their initial evaluation. At the same time, the reviewer is also responsible for judging the appropriateness of the methodology and accuracy of conclusions. These assessments can only be performed if the experiments are described in detail. Thus, we think the adoption of a reporting quality check by editor and reviewers, using the submission system and line references provided by the authors, could be implemented without a significant increase of the time necessary for review. We edited the text to reflect this position. It now reads “Journal editors and reviewers must actively enforce reporting guidelines for ongoing submissions as endorsing them does not improve compliance [47]. Promoting the adherence to higher reporting standards will also preseve the journal’s reputation. In the medium-term, journals can provide a list with relevant parameters (e.g. randomisation, blinding) with line references to be filled out by the authors in the submission system. Such a system would facilitate reporting quality checks by editors and reviewers without increasing the review length substantially. Registered reports allow input on the study design since they are submitted before the beginning of experiments. Additionally, since acceptance is granted before the results are known, the registered reports also encourage the publication of negative results, acting against the publication bias for positive outcomes.” L.354-62.
1.16 Lines 278-279: should read: "... within periodic updates of guidelines, ..."
We edited the text accordingly.
1.17 Line 280: should be: "... the strengths and weaknesses of validated disease models, ..."
We edited the text accordingly.
1.18 Lines 282- 319: This appears to be little more than a creditable wish-list. As such, I find it distracts from the message of the need to tighten the requirements for selection of models and reporting details. The authors should consider omitting much, if not all, of this paragraph. Overall, I suggest the authors think very clearly about the message for change that can be achieved, and then structure the paper accordingly. At present, it contains far too much listing of changes that could be made without identifying either timescales (short- versus long-term) or detailing the specific bodies (e.g. my comment re editors rather than reviewers) who should be responsible for the changes and, as such, the impulse to improve will be blunted.
We understand the reviewer apprehension about the presented stakeholder analysis diluting our main message. We advocate for tighter regulations for the selection of animal models and the performance of studies Indeed, some of our previous proposals could be seen as far-fetched and distract the reader from our goal. However, the change of culture in animal research is necessary, and therefore, this aspect must be considered in a discussion of animal model translatability. We have restructured the text to clarify our stakeholder analysis as well as to include more substantiation regarding the implementation. It now reads “Animal research is already a cost-intensive and often long endeavour. By conducting experiments with questionable validities, we are misusing animals – which is expressively prohibited by the European Union (EU) Directive 2010/63. Only with a joint effort involving researchers in academia and industry, ethics committees, funders, regulatory agencies and the pharmaceutical industry, we can improve the quality of animal research. By applying FIMD, systematic reviews and meta-analysis, and the IB-derisk, researchers can identify more predictive disease models, potentially preventing clinical trials starting based on unreliable data. These approaches can be implemented in the short-term in both the academic and industrial settings since the training requires only a few months.
Concomitantly, the other stakeholders must create an environment that encourages the adoption of best practices. Ethics committees have a unique opportunity to incentivise higher standards since an unfavourable assessment can prevent poorly designed experiments from even starting. However, they now frequently base their decisions on subpar efficacy data [26]. Also, the lack of a detailed disease model justification often results in the selection of disease models based on tradition, rather than science [47]. The request of a more detailed translational rationale for each model choice (e.g. by requiring models are evaluated with FIMD), as well as the enforcement of reporting guidelines, can act as gatekeepers for flawed study designs and improve the risk-benefit analysis significantly [48].
Funders can require the use of systematic reviews and meta-analyses and a thorough assessment of the translational relevance of selected animal models (e.g. FIMD). They can facilitate the adoption of these measures by reserving some budget specifically for training and the implementation of these strategies. Over time, FIMD, systematic reviews and meta-analyses can eventually become an essential requirement to acquire funding. Provided there is a grace period of at least a year, funders could request a stricter justification of model selection as early as their next grant round.
Journal editors and reviewers must actively enforce reporting guidelines for ongoing submissions as endorsing them does not improve compliance [49]. Promoting adherence to higher reporting standards will also preserve the journal’s reputation. In the medium-term, journals can provide a list with relevant parameters (e.g. randomisation, blinding) with line references to be filled out by the authors in the submission system. Such a system would facilitate reporting quality checks by editors and reviewers without increasing the review length substantially. Registered reports allow input on the study design since they are submitted before the beginning of experiments. Additionally, since acceptance is granted before the results are known, the registered reports also encourage the publication of negative results, acting against the publication bias for positive outcomes.
Regulatory agencies can shape the drug development landscape significantly with some key actions. For instance, updating the IB guidelines by requiring a more extensive translational rationale for each animal model employed would facilitate the risk-benefit analysis by assessors and ethics committees alike. This rationale should include not only an evaluation of the animal model itself but also how it compares to other available options. Furthermore, agencies could review disease-specific guidance to include a more comprehensive account of efficacy assessment by exploring the use of disease models in safety studies [24,50]. The simultaneous evaluation of efficacy and safety can result in more informative studies, which are more likely to translate to the clinic. Finally, the output of the translational assessments of animal models of disease can be incorporated within periodic updates of guidelines, for instance, as an extended version of Sheean and colleagues’ work [51]. Scientific advice can be used as a platform to discuss translational considerations early in development. By presenting the strengths and weaknesses of validated disease models, agencies can promote optimal model selection without precluding model optimisation and the development of new approaches.
Furthermore, drug development companies can significantly benefit from the implementation of these measures in the medium-term. Larger companies can perform a thorough assessment of preclinical data of internal and external assets using FIMD, systematic review and meta-analysis, and the IB-derisk. At the same time, small and medium enterprises can provide data in these formats to support their development plan. Ultimately, the selection of more predictive disease models will lead to more successful clinical trials, increasing the benefit and reducing the risks to patients, and lower development costs.
A positive side-effect of these strategies is the increased scrutiny of design and animal model choices. Instead of a status-quo based on tradition and replication of poor practices, we can move forward to an inquisitive and evidence-based modus operandi [7]. This change of culture is sorely needed in both academic and industrial institutions [52]. A shift toward a stricter approach – more similar to clinical trials, from beginning to end, is warranted [52].” L.331-418.
Reviewer 2 Report
General comments:
The manuscript “Levelling the animal to human translation gap in drug efficacy” describes three methods to assess the predictive value of animal models of disease. Since the subject matter is not my primary area of expertise, I mainly read the text from the viewpoint of a potential user of these methods ( a researcher working with animals models), but I may have missed some of the more technical questions that should be raised.
The “worked examples” for the FIMD (Diabetes type 2 and DMD) and the SRMA (effect of glucose-lowering agents) helped visualize how the tools work. Showing tha actual data of the examples, perhaps as a supplement, may be even more informative. The IB-derisk tool however does not really have such an example and (perhaps because of that) it is more difficult to understand what it actually does. I had to read the original article to be able to understand what the tool is and does. The explanation of how the IB-derisk tool works should be more clear so it is understandable for readers with less background in these matters.
Specific comments:
Line 97: You say there are 5 categories but only list 4
Line 131-134 what was the rationale for choosing these two models to validate the framework?
Line 153/table 1: Although all the criteria described here are certainly relevant, I feel like the choice of the model species itself remains somewhat overlooked. General characteristics of a species may not directly influence the modelling of the disease, but may influence the selected outcome parameters. For example, if pain behavior is one of the outcome parameters, the choice of species may very much influence how much of this behavior is actually displayed (prey animals tend not to display weakness due to the higher risk of predation). Or gastrointestinal peculiarities of the species (for example, absence of a gall bladder in some species, or the location of fermentation in the gut) may influence how certain medication works. All this seem very obvious considerations, but in practice I often see that the choice for a particular species is made based on tradition and convenience and not so much with general characteristics of the species in mind.
Line 225: An example of a sheet and a radar plot would be illustrative
Line 256-261: Do you think there is an additional role for editors and reviewers to promote the publication of negative results? Actively trying to counter publication bias could result in more realistic data input for your suggested tools.
Author Response
Reviewer #2: We thank the reviewer for their valuable input. A potential user is one of the most important stakeholders for the implementation of these methods.
General comments
The manuscript “Levelling the animal to human translation gap in drug efficacy” describes three methods to assess the predictive value of animal models of disease. Since the subject matter is not my primary area of expertise, I mainly read the text from the viewpoint of a potential user of these methods ( a researcher working with animals models), but I may have missed some of the more technical questions that should be raised.
2.1 The “worked examples” for the FIMD (Diabetes type 2 and DMD) and the SRMA (effect of glucose-lowering agents) helped visualize how the tools work. Showing the actual data of the examples, perhaps as a supplement, may be even more informative. The IB-derisk tool however does not really have such an example and (perhaps because of that) it is more difficult to understand what it actually does. I had to read the original article to be able to understand what the tool is and does. The explanation of how the IB-derisk tool works should be more clear so it is understandable for readers with less background in these matters.
We agree with the reviewer that the IB-derisk section is less detailed than the other two. We cannot include more results because the research is still ongoing, and it would prevent its publication. However, we edited the text to explain the IB-derisk tool in more detail. It now reads “The IB-derisk consists of a colour-coded excel sheet or web application (www.ib-derisk.org) in which PK and PD data can be inputted. It allows the extra- and interpolation of missing PK parameters across animal experiments, facilitating the dose selection in first-in-human trials.” L.271-4
Specific comments
2.2 Line 97: You say there are 5 categories but only list 4
We edited the text to include the fifth category (predictivity). It now reads “[…] consist of the simple scoring of five categories (species, disease simulation, face validity, complexity and predictivity) […]”. L. 100-1.
2.3 Line 131-134 what was the rationale for choosing these two models to validate the framework?
For type 2 diabetes, we chose the ZDF rat and the db/db mouse based on their extensive use in drug screening. For DMD, the mdx mouse was also chosen because of its extensive use, while the GRMD dog was selected for better replicating the human symptoms and histology. We edited the text accordingly. It now reads “To validate our framework, we first conducted a pilot study of two models of type 2 diabetes – the Zucker Diabetic Fatty (ZDF) rat and db/db mouse, chosen based on their extensive use in preclinical studies. Next, we did a complete validation of two models of Duchenne Muscular Dystrophy (DMD) – the mdx mouse and the Golden Retriever Muscular Dystrophy (GRMD) dog. We chose the mdx mouse owing to its common use as DMD model and the GRMD dog for its similarities to the human condition [31,32].” L.171-6.
2.4 Line 153/table 1: Although all the criteria described here are certainly relevant, I feel like the choice of the model species itself remains somewhat overlooked. General characteristics of a species may not directly influence the modelling of the disease, but may influence the selected outcome parameters. For example, if pain behavior is one of the outcome parameters, the choice of species may very much influence how much of this behavior is actually displayed (prey animals tend not to display weakness due to the higher risk of predation). Or gastrointestinal peculiarities of the species (for example, absence of a gall bladder in some species, or the location of fermentation in the gut) may influence how certain medication works. All this seem very obvious considerations, but in practice I often see that the choice for a particular species is made based on tradition and convenience and not so much with general characteristics of the species in mind.
We agree with the reviewer that model species has a substantial impact on disease outcomes. We demonstrated in previous research that indeed, tradition often prevails in animal model selection (Veening-Griffioen et al., 2020, ALTEX, doi: 10.14573/altex.2003301). Although the questionnaire in FIMD does not assess this specifically, the species should naturally be considered when answering each question. For instance, using the reviewer’s example, such considerations about pain behaviour would be described in the symptomatology and natural history domain. We have another example from the validation of the GRMD dog, in which we made considerations about mobility loss in the dogs (tetrapodal) when compared to humans (bipodal). Any relevant species-specific characteristic must be discussed within the applicable domain.
2.5 Line 225: An example of a sheet and a radar plot would be illustrative
While we agree with the reviewer some examples could be illustrative, the validation sheet is a dense document spanning ten pages on average (without references) per animal model. Also, it includes some features not discussed in depth in this paper. Since the four validation sheets are available online in FIMD’s original publication and are easily accessible (https://doi.org/10.1371/journal.pone.0218014.s004 and https://doi.org/10.1371/journal.pone.0218014.s005), we do not consider that it would add much value to this paper. However, we did add an example of a radar plot to illustrate FIMD’s output better. Figure 1, L.156-7.
2.6 Line 256-261: Do you think there is an additional role for editors and reviewers to promote the publication of negative results? Actively trying to counter publication bias could result in more realistic data input for your suggested tools.
Publication bias is an important topic, and we agree with the reviewer that lower publication bias could result in more reliable data to assess, validate and compare animal models of disease. There are already some journal initiatives addressing publication bias in research, such as the Journal of Negative Results. However, we think registered reports present an enormous opportunity since they allow acceptance before the results are known. This setting prevents non-acceptance based purely on negative results, leading to more accurate data. We edited the text to include the relevance of registered reports to that end. It now reads “Registered reports allow input on the study design since they are submitted before the beginning of experiments. Additionally, since acceptance is granted before the results are known, the registered reports also encourage the publication of negative results, acting against the publication bias for positive outcomes.” L. 359-62.
Reviewer 3 Report
This is an excellent short review in which the authors discuss their ideas of how to improve validity and translation of animal models of human disease. This is an important topic and I agree with the authors that implementing their ideas will improve translational research using animal models. I do not have any suggestions of how to improve the manuscript, except that the authors should consider avoiding the use of numerous uncommon abbreviations that make it difficult to follow the text.
Author Response
Reviewer #3: We thank the reviewer for their positive words.
This is an excellent short review in which the authors discuss their ideas of how to improve validity and translation of animal models of human disease. This is an important topic and I agree with the authors that implementing their ideas will improve translational research using animal models.
3.1 I do not have any suggestions of how to improve the manuscript, except that the authors should consider avoiding the use of numerous uncommon abbreviations that make it difficult to follow the text.
We have edited uncommon abbreviations, such as SRMA and DDC, to facilitate the reading.
Round 2
Reviewer 1 Report
Animals-836090-resubmission
I have read the resubmission thoroughly and compared it against the comments that I made in relation to the original. In my opinion, the authors have addressed most, but not all, of my earlier concerns. I am quite content that the authors choose to disagree with me; but some of my concerns required some sort of response and, as yet, the authors have not made any comment at all. For this reason, I cannot recommend acceptance just yet. I detail my comments below.
Lines 31-32: This addresses the point that I made in relation to Lines 149-151 of the original submission.
Lines 55-56: The point that I raised in relation to this sentence has not been addressed here; but is addressed properly at length much later (Lines 282-298). I am happy that in the way that it is addressed later; but I would suggest that the authors extend this sentence (Lines 55-56) to mention that such experiments mean that animals are unnecessarily subjected suffering.
Lines 115-121 & 126-131: This amendment deals with my concerns at Lines 115-116 in the original submission.
Lines 159-160: This amendment deals with my concerns expressed in relation to Line 137 of the original.
Lines 163-166: In the original submission at Lines 142-145, I asked the question as to whether or not these problems are surmountable. The authors have not responded. The problems might be: (i) definitely unsurmountable, in which case that avenue is closed; (ii) definitely surmountable, in which case the authors can suggest of future path; or (iii) unclear as to whether the problems are/are not surmountable.
I would, respectfully, suggest that the authors should not leave a question like this hanging without comment.
Lines 168-170: My concerns expressed with respect to Line 148 in the original have not been addressed. I suggest that the authors re-read my comments:
"Line 148: "to select optimal models". Of course, the best model available may still be a poor model. Given the issue raised at Lines 53-54, specifically experiments that are inherently flawed, can FIMD exclude models - even all the available models - to ensure unnecessary animal suffering is avoided?"
Line 181: The authors have not addressed my point relating to systematic reviews (Lines 169-170) in the original submission; they have simply removed any mention of the search term that they used. A cynic might be tempted to state that they have redrafted the paragraph to circumvent rather than address the comment.
Lines 187-189: This amendment deals with my comments in relation to Lines 165-166 in the original.
Lines 202-208: This amendment addresses my point in relation to Lines 175-178 of the original. However, the authors have not mentioned that this approach is not one that will lead to improvements in the short-term.
Lines 241-242: his deals with my comments in relation to Lines 216-218 in the original.
Lines 305-313: This addresses my concerns that I expressed at Lines 257-258 of the original.
Line 340: replace "shall" with "should"
Line 359: replace "bettered" with "improved".
Line 364: replace "clinic" with "clinical context".
In short, I think that the submission is much improved compared to the original. However, there is a little work that still remains to be done before it is ready for publication.
Author Response
Reviewer #1: We thank you again for your comments.
1.1. Lines 55-56: The point that I raised in relation to this sentence has not been addressed here; but is addressed properly at length much later (Lines 282-298). I am happy that in the way that it is addressed later; but I would suggest that the authors extend this sentence (Lines 55-56) to mention that such experiments mean that animals are unnecessarily subjected suffering.
We edited the text to include the reviewer’s suggestion. It now reads “Research has shown preclinical studies have major design flaws (e.g. low power, irrelevant endpoints), are poorly reported, or both – subjecting animal models to unnecessary suffering [6,7].” L55-7.
1.2 Lines 163-166: In the original submission at Lines 142-145, I asked the question as to whether or not these problems are surmountable. The authors have not responded. The problems might be: (i) definitely unsurmountable, in which case that avenue is closed; (ii) definitely surmountable, in which case the authors can suggest of future path; or (iii) unclear as to whether the problems are/are not surmountable. I would, respectfully, suggest that the authors should not leave a question like this hanging without comment.
We addressed this comment in the previous rebuttal on point 1.7, where we wrote “The study design and reporting deficiencies of animal studies undoubtedly represent a challenge to interpret the resulting data correctly. While owing to these deficiencies, many studies are less informative; some may still offer potential insights if the data are interpreted adequately. Assuming these shortcomings are insurmountable would mean dismissing a substantial learning opportunity. In clinical research, the conclusions from a randomised, double-blind trial are weighed differently than an open-label single-arm study. We need to apply the same standards to animal research to surmount the limitations of the current data. Concomitantly, we need to promote better study design, to generate more robust data and reporting quality, to differentiate between well and poorly designed studies accurately. These measures allow us to use available data while understanding and accounting for their deficiencies and encourage the implementation of better practices.” We added these considerations to the text. It now reads “Nonetheless, it presents challenges of its own, ranging from the definition of disease parameters and absence of a statistical model to support a more sensitive weighting and scoring system to the use of publicly available (and often biased) data. The latter is especially relevant, as study design and reporting deficiencies of animal studies undoubtedly represent a challenge to interpret the resulting data correctly. While owing to these deficiencies, many studies are less informative; some may still offer potential insights if the data are interpreted adequately. We included the reporting quality and risk of bias assessment to force researchers to account for these shortcomings when interpreting the data.” L180-189.
1.3 Lines 168-170: My concerns expressed with respect to Line 148 in the original have not been addressed. I suggest that the authors re-read my comments:
"Line 148: "to select optimal models". Of course, the best model available may still be a poor model. Given the issue raised at Lines 53-54, specifically experiments that are inherently flawed, can FIMD exclude models - even all the available models - to ensure unnecessary animal suffering is avoided?"
We addressed this point in the previous rebuttal (1.8). We wrote "The reviewer is completely correct. We designed FIMD to guide the assessment, validation and comparison of animal models of disease. As such, it is possible that the validation of all the models results in low scores, and thus, the inability to identify a relevant model. While the researcher may still choose to pick one of these models for a specific reason, FIMD also offers institutional review boards and funders a scientifically-sound rationale to refuse the performance of studies in poor animal models – even if it means no animal research should be conducted." We edited the reflect to include this consideration. It now reads “However, by itself, FIMD cannot prevent poor models from being used. The validation of all existing models may result in low scores, and thus, the inability to identify a relevant model. While the researcher may still choose to pick one of these models for a specific reason, FIMD offers institutional review boards and funders a scientifically-sound rationale to refuse the performance of studies in poor animal models – even if it means no animal research should be conducted.” L334-8.
1.4 Line 181: The authors have not addressed my point relating to systematic reviews (Lines 169-170) in the original submission; they have simply removed any mention of the search term that they used. A cynic might be tempted to state that they have redrafted the paragraph to circumvent rather than address the comment.
We addressed this comment in the previous rebuttal (1.11). We wrote "We agree that the only search term used is not optimal. Our point is that translational meta-analyses are still uncommon, and the paper by Leenaars and colleagues demonstrates it clearly. We edited the text to improve clarity. It now reads “The development of systematic reviews and meta-analyses to combine animal and human data offers an unprecedented opportunity to investigate the value of animal models of disease further. Nevertheless, translational meta-analyses are still uncommon [37]. A prospect for another application of systematic reviews and meta-analyses lies in comparing drug effect sizes in animals and humans directly.” L.220-4." We do not think an extensive review of translational meta-analyses will add value to our discussion on future methodologies. We edited the text to reflect the literature without conducting a mini-review. As discussed with the academic editor, we suggest to the keep the text as is.
1.5 Lines 202-208: This amendment addresses my point in relation to Lines 175-178 of the original. However, the authors have not mentioned that this approach is not one that will lead to improvements in the short-term.
We edited the text to include the lack of short-term results. It now reads “These translational tables, allied with more qualitative approaches, such as FIMD, could form the basis for evidence-based animal model selection in the future.” L225-6
1.6 Line 340: replace "shall" with "should"
We edited the text accordingly.
1.7 Line 359: replace "bettered" with "improved".
We edited the text accordingly.
1.8 Line 364: replace "clinic" with "clinical context".
We edited the text accordingly.